# Core-Shell Hydrogels with Tunable Stiffness for Breast Cancer Tissue Modelling in an Organ-on-Chip System

**DOI:** 10.3390/gels11050356

**Published:** 2025-05-13

**Authors:** Ilaria Parodi, Maria Elisabetta Federica Palamà, Donatella Di Lisa, Laura Pastorino, Alberto Lagazzo, Fabio Falleroni, Maurizio Aiello, Marco Massimo Fato, Silvia Scaglione

**Affiliations:** 1Department of Informatics, Bioengineering, Robotics, and System Engineering, University of Genoa, 16145 Genoa, Italy; donatella.dilisa@edu.unige.it (D.D.L.); laura.pastorino@unige.it (L.P.); marco.fato@unige.it (M.M.F.); 2National Research Council of Italy, Institute of Electronic, Computer and Telecommunications Engineering (CNR-IEIIT), 16149 Genoa, Italy; maurizio.aiello@cnr.it (M.A.); silvia.scaglione@cnr.it (S.S.); 3React4life S.p.A., 16152 Genoa, Italy; e.palama@react4life.com; 4IRCCS Ospedale Policlinico San Martino, 16131 Genoa, Italy; fabio.falleroni@hsanmartino.it; 5Department of Civil, Chemical and Environmental Engineering, University of Genoa, 16145 Genoa, Italy; alberto.lagazzo@unige.it

**Keywords:** hydrogels, tumour microenvironment, breast cancer, core–shell, cancer on chip

## Abstract

Breast cancer remains the most common malignancy in women, yet, many patients fail to achieve full remission despite significant advancements. This is largely due to tumour heterogeneity and the limitations of current experimental models in accurately replicating the complexity of in vivo tumour environment. In this study, we present a compartmentalised alginate hydrogel platform as an innovative in vitro tool for three-dimensional breast cancer cell culture. To mimic the heterogeneity of tumour tissues, we developed a core–shell structure (3.5% alginate core and 2% alginate shell) that mimic the stiffer, denser internal tumour matrix. The human triple-negative breast cancer cell line (MDA-MB-231) was embedded in core–shell alginate gels to assess viability, proliferation and hypoxic activity. Over one week, good cells proliferation and viability was observed, especially in the softer shell. Interestingly, cells within the stiffer core were more positive to hypoxic marker expression (HIF-1α) than those embedded in the shell, confirming the presence of a hypoxic niche, as observed in vivo. When cultured in the MIVO^®^ milli fluidic organ-on-chip resembling the physiological fluid flow conditions, cancer cells viability became comparable between core and shell hydrogel area, emphasising the importance of the fluid flow in nutrients diffusion within three-dimensional matrixes. Cisplatin chemotherapy treatment further highlighted these differences: under static conditions, cancer cell death was prominent in the softer shell, whereas cells in the stiffer core remained resistant to cisplatin. Conversely, drug diffusion was more homogeneous in the core–shell structured treated in the organ-on-chip, leading to a uniform reduction in cell viability. These findings suggest that integrating a compartmentalised core–shell cell laden alginate model with the millifluidic organ on chip offers a more physiologically relevant experimental approach to deepening cancer cell behaviour and drug response.

## 1. Introduction

Breast cancer poses a major global health challenge, being the most frequently diagnosed cancer and the leading cause of cancer-related mortality among women [1,2]. A critical factor driving cancer progression is the tumour microenvironment (TME), which influences crucial cellular processes such as proliferation, migration, invasion, and therapeutic resistance [3,4]. One of the key features of the TME is its heterogeneity, both in biochemical composition and mechanical properties, which together regulate the behaviour of cancer cells. With few exceptions, primary tumours are indeed significantly more rigid compared to the healthy tissues from which they originate, and this increased stiffness is strongly associated with higher malignancy [5]. Moreover, unlike healthy tissues, tumour tissues often exhibit stiffness irregularities, with stiffer regions typically located near the tumour core and softer regions towards the invasive front [6,7,8]. Furthermore, cells in stiffer matrices are more likely to undergo epithelial-to-mesenchymal transition (EMT), a critical process in metastasis, while cells in softer environments typically maintain a more proliferative and migratory phenotype [9,10].

In addition to the mechanical heterogeneity, solid tumours also suffer from poor vascularisation, which leads to nutrient and oxygen gradients within the tissue [11]. Hypoxia, or reduced oxygen availability, is a hallmark of aggressive tumours and occurs primarily in the central regions of the tumour mass, where cells are far from blood vessels. Hypoxia triggers the activation of hypoxia-inducible factor-1α (HIF-1α), a transcription factor that drives the expression of genes involved in angiogenesis, metabolic reprogramming, and survival [12,13]. This hypoxic response creates a selective pressure that enables the survival of more aggressive, therapy-resistant cancer cells [14]. Consequently, the combination of mechanical stress and hypoxia within the TME fosters the emergence of cancer cells with enhanced invasive potential and resistance to conventional treatments [15].

Given the crucial roles of both mechanical and hypoxic cues in tumour biology, there is increasing interest in developing in vitro models that can replicate these features to study cancer cell behaviour and test new therapeutic strategies. Conventional two-dimensional (2D) monolayer culture systems, although widely employed in the pharmaceutical industry for therapeutic agent development, fail to mimic the natural cell–cell and cell–matrix interactions. Additionally, cancer cells cultured on rigid 2D plastic surfaces exhibit changes in cellular behaviour, and drug response [16,17]. On the contrary, animal models, while valuable, present limitations in terms of human relevance and ethical considerations [18,19].

Therefore, three-dimensional (3D) hydrogel-based in vitro models have emerged as a promising solution for reproducing mechanical and biochemical features of the TME, such as tissue stiffness, cell/cell crosstalk, interaction with ECM and specific nutrient gradients [20]. For instance, collagen I-based and PEG-based hydrogels have been used to create tumour models, by reproducing hypoxic responses and by studying the effects of matrix stiffness on cancer cell behaviour [21,22]. However, many common 3D biomaterials, such as Matrigel, suffer from limitations including structural fragility and batch-to-batch variability, which can affect reproducibility in experimental outcomes [23,24].

In recent years, alginate has emerged as a versatile material for creating cancer models, thanks to its high biocompatibility, low-cost and tunable mechanical properties [25]. Alginate resembles indeed the glycosaminoglycans present in the native extracellular matrix (ECM), making this material highly appealing for many different applications [26]. Furthermore, the ionic gelation mechanism allows alginate to create a highly hydrated and cell-friendly microenvironment, closely resembling native ECM structures. Alginate hydrogels have been already employed to study the relationship between matrix stiffness and cancer cell viability [27]. They have also been used to develop tumour models for neuroblastoma, offering several advantages over traditional 2D cultures, including an in vivo-like expression of some of the most important immune-checkpoints [28], and an ovarian cancer model for drug efficacy tests, which showed drug response consistency with xenograft in vivo models [29].

In this study, a core–shell alginate hydrogel was developed to investigate breast cancer cell behaviour, particularly focusing on aggressive phenotypes. The hydrogel design features a stiffer inner core surrounded by a softer outer layer, mimicking the mechanical gradients observed in tumour tissues, where cells tend to proliferate and migrate more readily in softer regions [30,31]. Moreover, this core–shell structure can facilitate the formation of an oxygen gradient, typical of tumour tissue, inducing hypoxia in central regions [32,33]. Mechanical properties of the hydrogels, cell proliferation, and the induction of hypoxia were investigated, prior to investigate the impact of matrix stiffness of drug response to cisplatin—one of the most employed chemotherapeutic agents.

Recently, the combination of cells, spheroids or mini-tissues with microfluidic systems has opened new avenues in cancer research, ultimately leading to the emergence of tumour-on-a-chip models [34]. However, some fabrication techniques can be challenging to implement [35,36,37].

Moreover, aiming at simulating the perfused TME and systemic drug administration [38,39,40], this in vitro breast cancer model has been cultured in a multi-compartmental milli fluidic organ-on-chip to evaluate both the tumour cells viability and drug response under dynamic conditions. In parallel, computational studies have been performed to simulate the diffusive phenomena though the hydrogels and to calculate the shear stress perceived by the tissue models within a perfused organ-on-chip, aiming to optimise the 3D conditions. By leveraging this novel 3D platform, we aim to investigate the TME role in cancer progression and response to therapies, paving the way for more effective and personalised treatment strategies.

## 2. Results

### 2.1. Core–Shell Alginate Hydrogels Realisation and Multi-Level Characterisation

The design and fabrication of compartmentalised hydrogels are essential for mimicking complex tissue architectures and creating environments that can better simulate in vivo conditions. In particular, the development of core–shell hydrogels offers an opportunity to create gradients of stiffness and chemical composition, which can influence cell behaviour and drug delivery. In this study, compartmentalised hydrogels were obtained through a double agar mold casting method.

By using two different concentrations of alginate solutions, distinct core (3.5% *w*/*v*) and shell (2% *w*/*v*) layers were formed, with gelation driven by calcium ions in the agar molds at physiological temperature (T = 37 °C). Figure 1A illustrates the fabrication workflow and shows the resulting cell-laden models.

The core–shell hydrogels exhibited a well-defined structure, with a core measuring approximately 2 mm in diameter, encased within a shell. Visual inspection confirmed the integrity of the gels, with the core–shell hydrogels displaying a defined boundary between the core and shell layers (Figure 1B,C).

Morphology of the cell-free constructs was assessed through Scanning Electron Microscopy (SEM). Homogeneous constructs (2% *w*/*v* and 3.5% *w*/*v*) were also prepared and analysed for comparison. The microstructure of the different hydrogels (Figure 2) revealed that the pores within the soft gel (2% *w*/*v*) appeared relatively uniform in size and shape, while the stiff gels (3.5% *w*/*v*) showed a denser structure with fewer visible interconnected voids. At 500X magnification, the higher porosity of the softer construct was clearly noticeable compared to the stiffer one. Core–shell (CS) hydrogels presented distinct regions, with patterns similar to those found in both homogeneous constructs.

Furthermore, the hydrogels were mechanically analysed to obtain the stiffness values immediately after their production (Figure 3A). The mechanical properties of hydrogels play indeed a crucial role in determining their suitability for biomedical applications, as they directly influence cell behaviour, tissue integration, and the overall stability of 3D constructs. Understanding how these properties evolve over time is essential for designing materials that can mimic the physiological environment and support long-term functionality.

The measurements carried out through unconfined compression tests showed that core–shell hydrogels generally exhibit an intermediate stiffness between soft and stiff hydrogels. Moreover, cell-laden hydrogels were examined over a two-week culture period to assess the impact of breast cancer cells on the 3D constructs over time. Interestingly, the presence of cells affected hydrogel stiffness as early as one day into culture (Figure 3B). This immediate reduction in stiffness is likely due to a dilution effect. When cells are incorporated into the alginate gel, they occupy space that would otherwise be filled by the polymer, effectively reducing the local polymer concentration. This decrease in the density of the polymer network leads to fewer cross-links, which in turn diminishes the overall mechanical stiffness. However, despite this initial softening, the measured stiffness values remain within the physiological range [41,42].

Notably, while the stiffness remained nearly unchanged during the first week (Figure 3C), the hydrogels exhibited a progressive stiffening over the course of 14 days of cells culture (Figure 3D). This subsequent stiffening may be attributed to cellular remodelling and to the deposition of extracellular matrix components, which ultimately reinforce the structure of the hydrogel. All the data are summarised in Table 1.

Overall, these findings highlight that while the initial cell incorporation may dilute the gel and reduce its stiffness, the cellular activities over time lead to a remodelling process that restores and even enhances the mechanical integrity of the 3D constructs.

### 2.2. Core–Shell Structure Shows a Hypoxic Core While Sustaining Cancer Cells Proliferation

The interaction between breast cancer cells and the surrounding hydrogel matrix is critical for understanding how these constructs support cell viability, proliferation, and function. In this study, breast cancer cells cultured in alginate-based core–shell hydrogels were characterised by immunostaining for their morphology.

Cells displayed a round morphology and a homogeneous spatial distribution within both the 2% shell and 3.5% core alginate matrices, with no significant differences between the two compartments.

In addition, cells demonstrated a robust proliferation, as evidenced by the Ki67-positive signal, and cell density significantly increased after one week. Notably, immunofluorescence revealed colocalisation of Ki67 and HIF-1α, particularly in the stiffer core regions (Figure 4). This result suggests that hypoxic conditions can be recapitulated in vitro by finely tuning the stiffness of the extracellular matrix where cells are embedded: indeed, as in vivo, the density and mechanical properties of the ECM directly affect the oxygen diffusion, which finally leads to a non-homogeneous expression of cellular hypoxic marker.

### 2.3. The Core–Shell Hydrogel Cultured Within a Millifluidic Organ on Chip Displays an Increasing of Cells Viability in the Stiffer Core Compared to the Static Culture

To further reproduce the in vivo physiological environment, core–shell hydrogels were cultured in an innovative organ-on-chip platform, MIVO^®^, to assess their behaviour when exposed to physiological fluid flow. As control, the traditional static culture was used.

Initially, cells viability was assessed with a Live/Dead assay to determine whether the nutrient delivery in dynamic systems could mitigate the limitations of denser, stiffer regions within the hydrogel and improve cell viability (Figure 5).

Interestingly, in static conditions, the cell viability in core–shell hydrogels was approximately 77% and 91% in the core and in the shell, respectively, showing that a softer matrix might offer a more suitable substrate for the breast cancer cells growth, in agreement with previous findings by Cavo et al. [27]. When the core–shell hydrogels were cultured in the organ-on-chip under physiological fluid flow conditions, cell viability in the stiffer core region increased significantly, reaching levels comparable to those in the softer shell (92%). This result can be attributable to the enhanced diffusion of nutrients, such as glucose, under dynamic culture conditions, which might mitigate the limitations imposed by the denser matrix and improve cellular survival in stiffer regions.

These findings are consistent with the study by Marrella et al. [29], which emphasised the importance of nutrient diffusion in 3D cell cultures under fluid flow. The improved viability under dynamic conditions highlights the potential of using organ-on-chip platforms to better replicate in vivo-like environments, where both mechanical properties and nutrient transport are critical for cell survival and behaviour.

### 2.4. Computational Analysis

To further highlight the crucial role of fluid flow, especially in 3D cell cultures, a computational simulation of nutrient (i.e., glucose) and oxygen distribution was conducted both in the presence and absence of flow.

For the static culture simulation, a geometry comprising the gel immersed in a 24-well plate was considered, exploiting an axisymmetric structure (Figure 6A). In this case, mass transport takes place only via diffusion. Differently, for the culture in the organ-on-chip, fluid dynamics within a Single-Flow MIVO^®^ device (Figure 6B) was first investigated, showing a tumour-like interstitial flow under the gel inserted on the transwell insert, when the imposed inlet flow rate is 1 mL/min [6]. Then, the transport was calculated, taking into account the velocity profile previously obtained.

A gradient of nutrient was found in static condition for oxygen and glucose. In particular, after 24 h, the oxygen decreased at the bottom of the construct up to 0.04 mM that is considered a level close to hypoxia for most cells (Figure 6C,D). While cancer cells preferentially convert glucose to lactate even when oxygen is available, our simulations showed that glucose concentrations did not decrease significantly over time (Figure 6G,H).

Under dynamic conditions, oxygen levels were higher compared to static conditions, with a more uniform distribution throughout the hydrogel. As shown in Figure 6E,F, the top and bottom sections of the scaffold exhibited similar oxygen levels, likely due to the continuous oxygen supply at the medium-air interface and the diffusion of oxygen through the silicone tubing of the millifluidic system. In terms of glucose distribution, the nutrient did not show a significant gradient after 24 h (Figure 6I,J). A slightly higher concentration was observed at the bottom of the hydrogel, probably due to more effective glucose delivery thanks to the fluidic flow. These results indicate that dynamic flow minimises the formation of concentration gradients, ensuring that cells receive a uniform supply of nutrients and oxygen.

### 2.5. The Administration of Drug Within the Organ on Chip Leads to a More Homogeneous Cytotoxicity Effect on the Core–Shell Cell Laden Hydrogels

Cisplatin is a widely used chemotherapeutic agent known for its ability to induce DNA damage and inhibit cell proliferation, making it a cornerstone of cancer treatment. However, available preclinical in vitro models do not reflect the proper in vivo drug penetration and cells response. On the other side, animal models, although properly resembling the drug pharmacodynamics, are not fully humanised models, and they have species-specific differences, leading to unpredictive readouts.

To overcome these limitations, we assessed the impact of cisplatin-induced cytotoxicity by using a fluid-dynamic organ on chip, which recapitulates the human circulatory flow. By using a core–shell alginate hydrogel system cultured in the MIVO^®^ organ-on-chip platform, we aimed to simulate the heterogeneous environment of a tumour, where the stiffness gradient could potentially affect drug diffusion and cellular response. The impact of cisplatin-induced cytotoxicity was assessed under fluid dynamic conditions, by applying the drug in the basal circuit of the organ on chip, while the core–shell alginates were cultured in the apical chamber. Static drug administration was used as internal control.

Under static conditions, a reduction in cell viability was observed only in the softer external area, which was more exposed to the drug. Conversely, in the internal, stiffer region, cells viability remained significantly higher (over 80%), suggesting limited drug penetration. Interestingly, when the same gels were treated with cisplatin within the MIVO^®^ organ on chip platform under fluid flow conditions, cell viability decreased uniformly across both the core and shell, confirming the crucial role of the flow in ensuring proper drug diffusion throughout the tumour matrix (Figure 7). Moreover, under dynamic conditions, the overall cell viability was approximately 60%, representing an intermediate value between the viability observed in the static core (45%) and static shell (84%). This suggests that, in static conditions, drug molecules accumulate more in the outer shell, leading to a higher cytotoxic effect compared to the more homogeneous distribution achieved under dynamic flow.

Furthermore, the cisplatin concentration was computationally simulated throughout the entire cell culture period, starting with an initial concentration of 10 µM in the medium. Two different points were considered: one located along the circumference of the top surface (A), representing the shell, and the other at the centre of the bottom surface (B), representing the core. The simulations revealed a plateau after approximately 4–5 h, indicating rapid diffusion of the drug, combined with slow consumption by the cells, which led to drug accumulation within the gel over time (Figure 8).

In the organ-on-chip system, higher drug transport within the hydrogel compared to the static culture was observed, in accordance with previous studies conducted on this perfused system [29,43].

## 3. Discussion

The creation of three-dimensional (3D) hydrogel models that closely mimic the in vivo tumour environment is a critical area of research, particularly for cancer cell biology, and drug delivery applications. Hydrogels, as biomaterials, can replicate the structural and mechanical properties of cancer tissues more accurately than traditional 2D cultures. Among the various hydrogel designs, core–shell structures have gained considerable attention due to their ability to mimic the heterogeneous microenvironments of tumours, where stiffness gradients exist between the tumour core and the outer shell.

Tumours in vivo exhibit indeed a highly complex microenvironment characterised by both biochemical and biophysical heterogeneity. The mechanical properties of tumours, such as stiffness, vary greatly between the tumour core and the peripheral regions, primarily due to factors like extracellular matrix (ECM) composition, cell density, and vascularisation [44]. The core of a solid tumour is typically stiffer and more hypoxic, while the periphery is softer and better oxygenated, providing a spatial heterogenic environment that may significantly affect cancer cellular behaviour, including growth, apoptosis, and response to drugs [45].

Recently researchers developed 3D models that mimic tumour stiffness heterogeneity based on gradient-based hydrogels or intricate compartmentalisation methods, which can pose challenges in terms of scalability and consistency [46,47,48,49].

This study presents a novel simple approach using compartmentalised core–shell hydrogels, to simulate tumour-like features by varying the stiffness between the core (3.5% *w*/*v* alginate) and shell (2% *w*/*v* alginate) regions. The strategy of embedding the CaCl_2_ solution within the agar mold, to crosslink the different alginate solutions, was found to be critical for ensuring the structural fidelity of the core–shell constructs. External cross-linking approaches tested during preliminary experiments resulted in uneven gelation and inconsistent shell formation, which adversely affected reproducibility. In contrast, the agar-embedded CaCl_2_ provided a spatially confined and stable ionic source that enabled gradual and uniform cross-linking via diffusion. This configuration supported more consistent shell architecture and homogeneous core encapsulation, likely contributing to the improved mechanical uniformity observed across samples. Tumour cells in softer, more compliant regions are indeed generally more proliferative, whereas stiffer regions may induce quiescence or apoptosis [50]. The study highlighted that softer 2% *w*/*v* alginate hydrogel exhibited a more open and uniform pore structure, which aligns with findings in the literature, reporting that softer gels generally exhibit higher porosity and an improved permeability to nutrients and oxygen. Conversely, the stiffer 3.5% *w*/*v* alginate gel exhibited a denser structure, which can lead to reduced diffusion of nutrients and waste, closely mimicking the poor nutrient perfusion found in the hypoxic core of tumours.

In addition, our core–shell cell laden hydrogels displayed different mechanical properties (i.e., Young modulus), with values recapitulating the physiological cancer stiffness. Importantly, the stiffer core region was correlated to higher HIF-1α expression, replicating a key feature of the tumour microenvironment.

The absence of marked morphological differentiation between the core and shell regions suggests that the cells maintained a uniformly round morphology and consistent spatial distribution throughout the hydrogel. This uniformity may be ascribed to the limited culture duration, which was likely insufficient to permit substantial cytoskeletal remodelling and active cell migration.

Computational analysis further supported the experimental readouts, by showing a different nutrient transport (oxygen and glucose) in static versus dynamic cell cultures. While in static conditions oxygen drops to levels close to hypoxia, a more homogeneous distribution was indeed achieved in dynamic conditions. Although not addressed in the present work, continuous monitoring of nutrient levels would undoubtedly provide a more comprehensive understanding of the cellular microenvironment, particularly in the context of extended 3D culture conditions.

These core–shell hydrogels resembling the heterogenous tumour microenvironment were then cultured within an advanced organ on chip platform, where a circulatory environment was adopted to deliver both nutrients and drug treatments, under physiological capillary velocities, to mimic the systemic anticancer drug administration. The growing need to test new chemotherapeutic drugs and to better understand the behaviour of tumours has driven the development of more sophisticated in vitro models, capable of accurately replicating tumour conditions in a fluid flow environment and providing critical information on their effectiveness.

In particular, we observed the effects of fluid flow on drug diffusion and the subsequent cellular response in the core–shell hydrogel system. When cultured under dynamic fluid flow conditions, the cancer cell viability decreased uniformly across both the core and shell regions, suggesting that the fluid flow facilitated more homogeneous and physiological drug distribution, unlike the static conditions where the absence of flow determined a non-physiological drug accumulation in the outer shell, leading to non-reliable high cytotoxic effect. This result is consistent with previous studies demonstrating that fluid flow can enhance nutrient and drug diffusion within 3D hydrogel cultures, mitigating the concentration gradients often observed in static conditions [51]. Moreover, the induced cytotoxicity observed within the organ on chip was in line with results obtained with the same drug dose in a xenograft model of ovarian cancer [27].

The cisplatin cytotoxicity experiments in this study further emphasise the importance of fluid flow. In particular a validated experimental approach by Marrella et al. [29] was employed. Under static conditions, the softer outer shell of the hydrogels, being more accessible to the drug, exhibited a greater reduction in cell viability compared to the stiffer core. In contrast, when the system was cultured under dynamic flow, the drug was more evenly distributed, leading to a more uniform decrease in cell viability. These findings align with the work of Cavo et al. [23], who showed that fluid flow within organ-on-chip models enhances drug delivery and penetration, improving the therapeutic response of cancer cells. Additionally, computational simulations conducted in this study confirmed the faster and more uniform diffusion of cisplatin under flow conditions, further supporting the beneficial role of fluid perfusion in 3D cultures [52]. Taken together, these results show that the core–shell hydrogel system offers a promising model for simulating the complex mechanical and biological environments of tumours. By combining distinct stiffness gradients and fluid flow, this model provides a more accurate representation of the tumour microenvironment, which is critical for understanding cancer biology and improving therapeutic strategies.

## 4. Conclusions

The results from this study highlight the potential of core–shell hydrogels for mimicking the mechanical gradients found in tumours. The ability to create a stiffness gradient within a 3D hydrogel allows for the study of cellular behaviour in environments that closely resemble those encountered in vivo, offering insights into tumour progression, metastasis, and drug resistance mechanisms. Most current 3D models that replicate tumour stiffness heterogeneity rely on gradient hydrogels or complex compartmentalisation techniques, which often limit scalability and reproducibility. In contrast, our core–shell hydrogel system offers a simple, modular, and reproducible platform with clearly defined geometry and mechanically distinct regions. The implemented stiffness contrast effectively mimics the mechanical heterogeneity found in solid tumours, such as breast cancer, where stiff, hypoxic cores are surrounded by softer peripheral regions.

By using the same cell type and density in both compartments, the model isolates the effect of matrix stiffness on cell behaviour. Biologically, it captures key tumour features, including spatial hypoxia patterns and differential cell viability, potentially reflecting the emergence of more aggressive subpopulations within the stiffer, hypoxic core.

Moreover, the integration of fluid flow within the organ-on-chip platform provides a more realistic environment for drug diffusion and cellular responses, addressing the limitations of traditional 2D and static 3D cultures. This study further supports the growing body of evidence that tumour models with mechanical and fluid flow considerations are crucial for more accurate drug testing and the development of effective therapies.

While the dynamic monitoring of nutrient and metabolite gradients was beyond the scope of the present work, such aspects remain critical for a more comprehensive understanding of long-term cellular responses. The integration of metabolic assays or real-time biosensors in future studies could provide valuable insights into the evolving microenvironment, further enhancing the physiological relevance of in vitro tumour models.

Future work could also explore the incorporation of additional features, such as cellular heterogeneity and ECM components, to further enhance the physiological relevance of these models for cancer research. In particular, future studies could investigate cell behaviour over a longer period and observe potential migration out of the hydrogel to assess their metastatic potential. Additionally, tumour microenvironment cells, such as fibroblasts, could be incorporated into the shell to examine their impact on cancer cells and their drug resistance, or endothelial cells could be included to vascularise the model.

Finally, the integration of PK modelling in future developments could improve the physiological relevance of the system by simulating drug exposure profiles and flow-mediated gradients, offering deeper insights into drug distribution dynamics and enabling more accurate predictions of therapeutic responses in relation to tumour microenvironmental features such as stiffness and hypoxia.

## 5. Materials and Methods

### 5.1. Hydrogels Fabrication

A 1% (*w*/*v*) agar solution was prepared by dissolving agar powder (Sigma, CAS 9002-18-0, Kanagawa, Japan) in a 0.1 M CaCl_2_ solution, representing the alginate cross-linker (Riedel-de Haen, CAS: 10043-52-4, Seelze, Germany). The mixture was heated to boiling, then poured into Petri dishes to a height of approximately 1.5–2 mm. After cooling and complete solidification, 2 mm and 5 mm diameter holes were created in the agar using biopsy punches to form molds. To prepare alginate solutions, alginate powder (Sigma, CAS: 9005-38-3) was mixed in physiological buffer (0.9% NaCl) at concentrations of 2% and 3.5% (*w*/*v*). The solutions were sterilised in an autoclave at 121 °C.

Agar molds containing CaCl_2_ were used to allow the gelation of the alginate solutions at a physiological temperature (T = 37 °C). First, the 3.5% solution was poured in the 2 mm diameter molds, to create the core structure. After 30 min, the resulting gels were gently removed with tweezers and transferred to the larger molds, where the 2% solution was dispensed to form the shell. After further 30 min crosslinking at T = 37 °C, the core–shell hydrogels were obtained. Homogeneous alginate gels were also fabricated, by pouring either the 2% or 3.5% solutions into 5 mm diameter molds and allowing them to crosslink for one hour. The 3D constructs were then removed from the agar and placed in a 5 mM CaCl_2_ maintenance solution. Preliminary tests showed that alternative cross-linking strategies, such as dripping CaCl_2_ onto alginate preloaded in PDMS molds, led to uneven gelation fronts, inconsistent shell thicknesses, and reduced structural fidelity.

### 5.2. Scanning Electron Microscopy

SEM analysis of the alginate samples was conducted using a Hitachi scanning electron microscope, model S-2500. The hydrogels were immersed in liquid nitrogen for 10–15 min, then transferred into a lyophilizer for 24 h to initiate a dehydration process. Then, the samples were coated with a thin layer of gold, employing the Sputtering system Polaron, and then observed at 200× and 500× in secondary electron, operating at a voltage of 10 kV. The obtained pictures were analysed with ImageJ (v1.54d).

### 5.3. Cell Culture

Commercially obtained human MDA-MB-231 cells, an adherent cell line derived from the pleural effusion of a primary triple-negative breast adenocarcinoma, were used in this study. Cells were expanded in DMEM medium supplemented with 10% Fetal Bovine Serum (FBS), 1% penicillin-streptomycin (P/S) and 1% L-glutamine (all from Sigma Aldrich). Once the cells reached confluency and the desired number was obtained, the MDA-MB-231 cells were enzymatically detached using 0.05% trypsin and subsequently counted. The cells were then suspended in alginate solutions at a concentration of 10^6^ cells/mL and poured into agarose molds as described previously to form the gels. The resulting hydrogels were cultured in 24-well plates for up to 7 or 14 days, using the complete medium supplemented with 5 mM CaCl_2_. The cell-laden hydrogels were cultured in incubator at controlled conditions (i.e., 5% CO_2_ atmosphere, temperature of 37 °C).

### 5.4. Mechanical Analysis

The characterisation of mechanical properties of alginate scaffolds was performed with a uniaxial testing machine (Z1.0, ZwickRoell, GmbH & Co. KG, Ulm, Germany) at room temperature with a speed of 1 mm/min. The compression measurements were carried out up to the 50% of the load. All measurements were performed as quadrupled and analysed with the software testXpert II. The compressive Young modulus *E* was calculated within the linear range of the slope (ε = 0–20%) of the obtained technical stress–strain curves:$$E=\frac{\sigma }{\epsilon }=\frac{Fl_{0}}{A∆l}$$
where *A* is the idealised cross-section and *l_0_* length of the uncompressed samples, ∆*l* the technical change in length and *F* the nominal force. Both the cell-free and the cell-laden hydrogels were considered. For the 3D construct with embedded cancer cells, the Young modulus was investigated after 24 h, 7 days, and 14 days of culture.

### 5.5. Drug Testing and Dynamic Culture

The MIVO^®^ organ on chip device was used as diffusive culture chamber able to culture the hydrogel-based breast cancer models under physiological flow conditions, providing a fluidic circulation below the tissues mimicking the human circulatory system. The receiver compartment was connected to a peristaltic pump, able to induce a monodirectional flow at capillary velocities (1 mL/min flow rate). The cisplatin chemotherapeutic agent was used to test its cytotoxicity in vitro. The drug was added to the culture medium in the fluidic circuit, mimicking the systemic drug administration, at a final concentration of 10 μM, that is close to the plasmatic concentration measured in mice models [29]. Similar cell laden hydrogels were treated under static conditions, as control. Untreated samples were also included, as negative control. The experiment was conducted for 7 days, and the culture medium was changed after 3 days, readministering 10 μM of cisplatin to the treated gels.

### 5.6. Cell Viability

The viability of MDA-MB-231 cells embedded within alginate gels was investigated through live/dead assay. Briefly, after 7 days of treatment, samples were washed with a physiological buffer (0.9% NaCl) with 5 mM CaCl_2_ and incubated in 0.1 μM Calcein-AM (Thermofisher, Waltham, MA, USA) and 0.3 μM Propidium Iodide (PI) (Sigma) in the buffer solution for 1 h at 37 °C in a dark environment to detect live and dead cells, respectively. Then, samples were washed three times in buffer solution and observed under a fluorescence microscope (Nikon H550L, Shinagawa, Japan). Pictures were and processed with ImageJ^®^ software. Cell viability was derived as the ratio between the number of alive cells and the total number of cells for each picture.

### 5.7. Immunostaining

Markers of cytoskeleton (phalloidin), proliferation (Ki67) and hypoxia (HIF-1 alpha) were also investigated in the cell-laden hydrogels. After the three-time washing step, hydrogels were fixed with 4% paraformaldehyde (PFA) and incubated in a permeabilisation solution (0.1% Triton X-100) at room temperature for 2 h. Subsequently, hydrogels were incubated with 2% bovine serum albumin (BSA) for 2 h.

The primary antibodies (Abcam, Cambridge, UK) Rabbit Anti-ki67 and Mouse Anti-HIF1α were diluted in 0.2% BSA at the concentration of 1:500 and 1:200, respectively, and employed to marker the cells in the alginate hydrogels. The incubation took place overnight at T = 37 °C to improve the penetration of the markers. Subsequently, the hydrogels were incubated for 2 h with secondary antibodies (Abcam) Anti-rabbit 488 and Anti-mouse 555 both, diluted 1:200 in 0.2% BSA.

For the cytoskeleton visualisation samples were incubated with Phalloidin Fluorescein Isothiocyanate Labeled 1:40 in the buffer, after the BSA for 1 h. Then, cells nuclei were counterstained with DAPI diluted 1:5000 in the buffer. Pictures were taken through a Nikon Eclipse Ti2-E. Confocal microscope (Olympus IX-81, Tokyo, Japan) was also employed to obtain a three-dimensional reconstruction of the hydrogels.

### 5.8. Computational Analysis

A computational analysis of the fluid dynamics recapitulated within the organ on chip device was carried out to predict the fluid velocity and shear stress perceived by cell-laden hydrogel on a commercial 24-well insert. The study was carried out using the Free and Porous Media Flow module of Comsol Multiphysics 6.0, with the following considerations: (1) a stationary and laminar flow, (2) an incompressible Newtonian fluid, and (3) the 24-well insert membrane as a porous medium. Hence, the Navier–Stokes Equation (1) and the Continuity Equation (2) for mass conservation were applied:(1)$$\rho \left(u\cdot \nabla u\right)=-\nabla p+\mu \nabla^{2}u$$(2)$$\rho \left(\nabla \cdot u\right)=0$$
where ***u*** is the velocity and *p* the pressure across the circuit. The culture medium density *ρ* and dynamic viscosity *μ* values were for water at room temperature. As initial conditions, the velocity field and the pressure were considered null. It was imposed a flow rate of 1 mL/min to induce fluid motion as input, according to the experimental set up, while a zero-pressure was set as output, avoiding the backflow. Additionally, a no-slip boundary condition was applied to the wall of the organ-on-chip. Furthermore, the Darcy’s law Equation (3) was considered for the 24-well insert membrane:(3)$$u=-\frac{\kappa }{\mu }\nabla p$$
where *κ* (m^2^) is the permeability of the membrane, which depends on the porosity *ε_p_* and on the diameter of the pores *d* according to the following expression(4)$$\kappa =\varepsilon_{p}\frac{d^{2}}{32}$$

The porosity of 13% was determined by considering the pores diameter (0.4 μm) and pore density (10^8^/cm^2^) according to the commercial insert characteristics.

Subsequently, oxygen, glucose and cisplatin mass transports were investigated during the first 72 h of culture through the Transport of the diluted species (TDS) module of Comsol Multiphysics 6.0. The general form to describe mass transport of a component *s* can be written as follows:(5)$$\frac{\partial s}{\partial t}+\nabla \cdot (-D\nabla s)+u=R$$
where the diffusive term is coupled with a convection transport, due to the presence of a velocity field (***u***), and the metabolite and drug consumption due to cellular activity. The different zones of the hydrogel are characterised by two different diffusivities for the nutrients and the cisplatin. It has been chosen to model the diffusion coefficient in the two domains as *D_soft_* = 1.3 *D_stiff_* interpolating the values found in the study of Hust et al. [53]. The reaction term *R* for the nutrient/drug was defined according to the Michaelis–Menten kinetics:(6)$$R=V_{max}\frac{sn}{s+K_{m}}$$
where *s* is the concentration of the species, *V_max_* represents the maximum consumption rate, and *K_m_* represents the component concentration when the rate is *V_max_*/2. The initial oxygen concentration is 0.2 mM everywhere. The glucose initial concentration in the culture medium DMEM is 25 mM. The cisplatin initial concentration in the circuit is 10 µM, while no drug is present in the upper chamber of the organ-on-chip, where the hydrogel is put. A constant oxygen boundary condition of 0.2 mM was set at the interface between the culture medium and the air and on the external wall of the silicon circuit pipes.

The study was also conducted for the static cell laden hydrogel condition, neglecting the velocity field and considering the hydrogel put in a 24-well culture system, exploiting an axisymmetric geometry.

An iterative geometric multigrid (GMRES) algorithm was employed to solve the equations for the steady-state condition. MATLAB R2024b was used to process the data. The other parameters used for the simulations are reported in Table 2.

### 5.9. Statistical Analysis

Data were analysed with Matlab software. Two-way ANOVA was used for viability of MDA-MB-231 cells comparing the two concentration of alginate and the two conditions of culture. Level of significance was set at *p* < 0.05 (* *p* < 0.05, ** *p* < 0.01). Number of replicates for each experiment are reported in figure legends.

## Figures and Tables

**Figure 1 gels-11-00356-f001:**
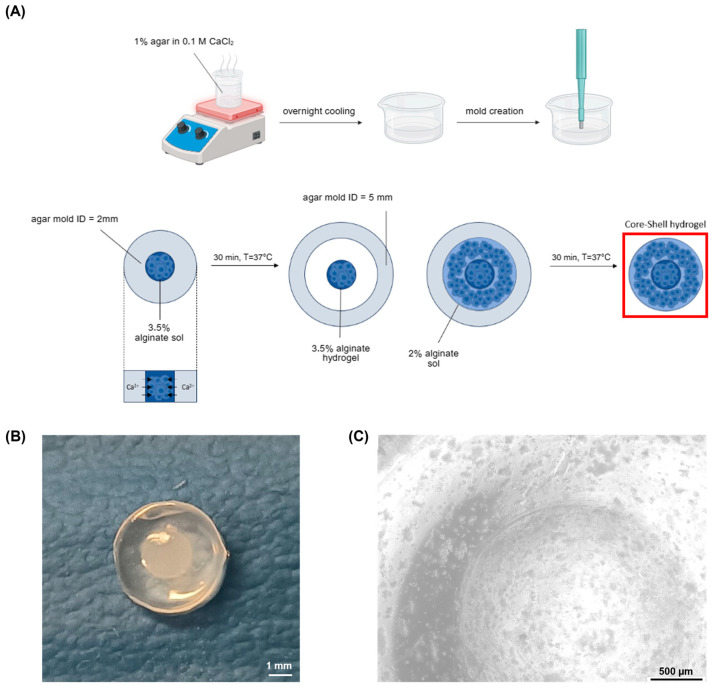
Core–shell hydrogel development (**A**) cell-laden core–shell hydrogel fabrication workflow: first, CaCl_2_-laden agar gels were produced, boiling the agar dissolved in 0.1 M CaCl_2_ solution, cooling it in Petri dishes and creating holes with 2 mm and 5 mm diameter biopsy punches; then the cell-laden hydrogels were realised with two subsequent gelation processes, each one with a duration of 30 min, performed at physiological temperature (**B**) cell-free core–shell hydrogel structure. Scale bar: 1 mm (**C**) Microscope image of the cell-laden core shell hydrogel. Scale bar: 500 µm.

**Figure 2 gels-11-00356-f002:**
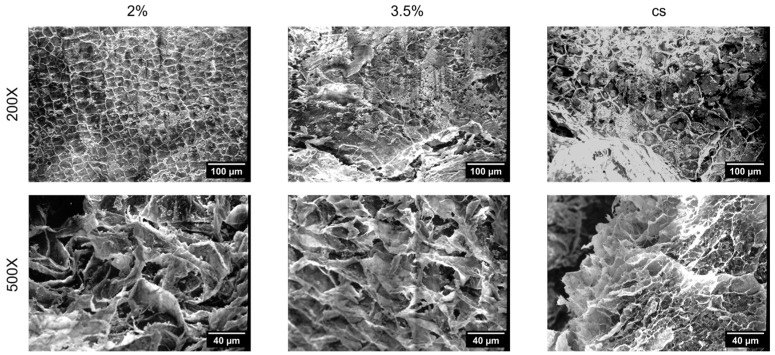
Scanning electron microscopy (SEM) analysis. SEM micrographs of the hydrogels produced with the agar mold technique at different magnification: 200× (scale bar: 100 µm) and 500× (scale bar: 40 µm). CS: core–shell. Homogeneous hydrogels fabricated using single-concentration alginate solutions (2% and 3.5%) were also realised for comparison.

**Figure 3 gels-11-00356-f003:**
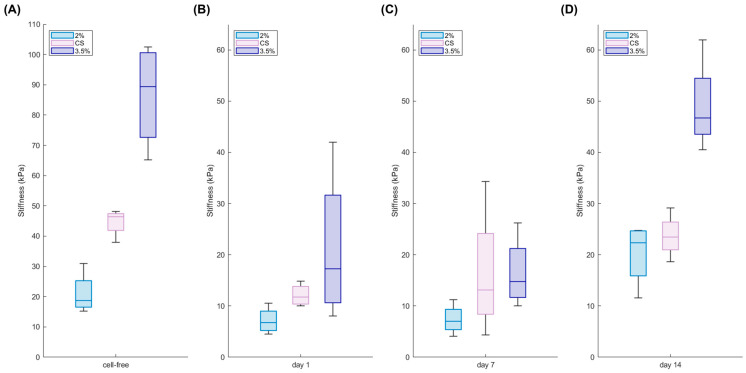
Mechanical properties of cell-free and cell-laden hydrogels. Young modulus of the core–shell hydrogels (CS) in comparison with soft (2%) and stiff (3.5%) homogeneous hydrogels during 14 days of culture (bottom): analysis carried out on cell-free hydrogels (**A**) and on cell-laden hydrogels at day 1 (**B**), day 7 (**C**), and day 14 (**D**) (*n* = 4 replicates).

**Figure 4 gels-11-00356-f004:**
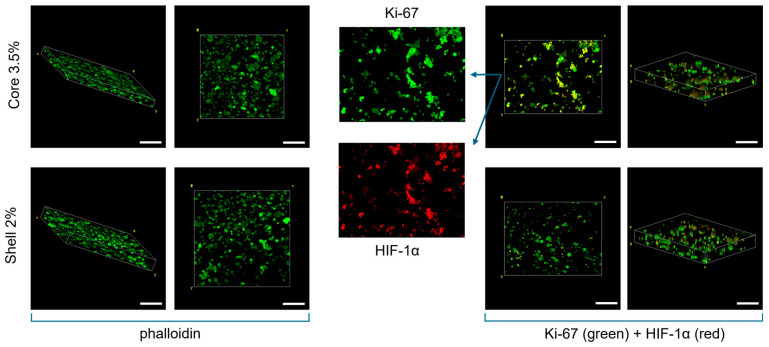
Confocal Immunofluorescence images. Phalloidin staining (**left**) and Ki67 and HIF-1α immunostaining (**right**) of the core–shell hydrogels after one week of culture. Globular morphology was investigated through the actin-related marker. Co-localisation of Ki67 and HIF-1α is highlighted, with more evidence in the core part of the 3D construct. Scale bar: 200 µm.

**Figure 5 gels-11-00356-f005:**
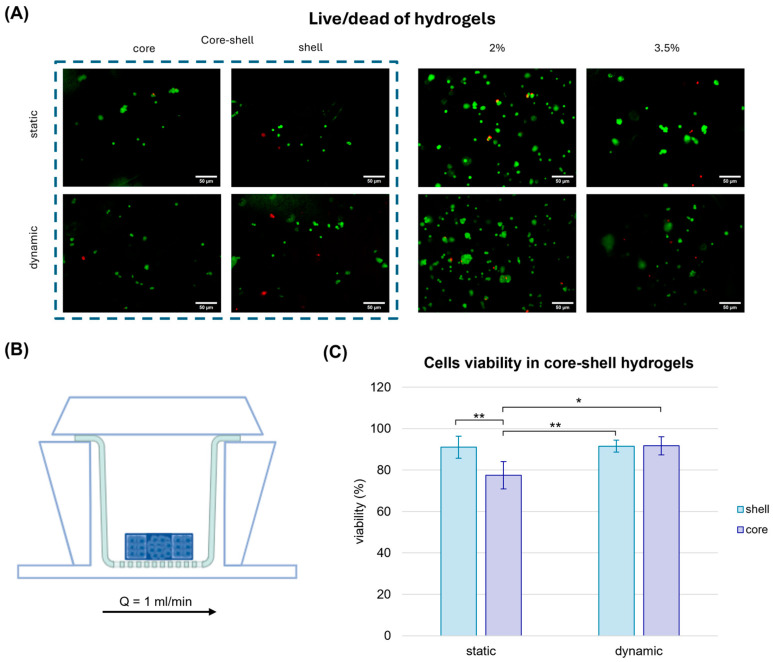
Cell viability of cell-laden hydrogels in static and dynamic conditions. Cell viability assay performed in alginate hydrogels after one week of static culture and dynamic culture in MIVO^®^ organ-on-chip. (**A**) Live cells were marked with Calcein-AM, dead cells were marked with Propidium Iodide. Scale bar: 50 µm. (**B**) MIVO^®^ device with the hydrogels located over a transwell insert subjected to a perfusion of 1 mL/min. (**C**) Cell viability evaluation in the core–shell hydrogels. The values are represented by mean ± std (*n* = 5). Statistics performed with two-way ANOVA; significance was assumed for *p* < 0.05 (* *p* < 0.05, ** *p* < 0.01).

**Figure 6 gels-11-00356-f006:**
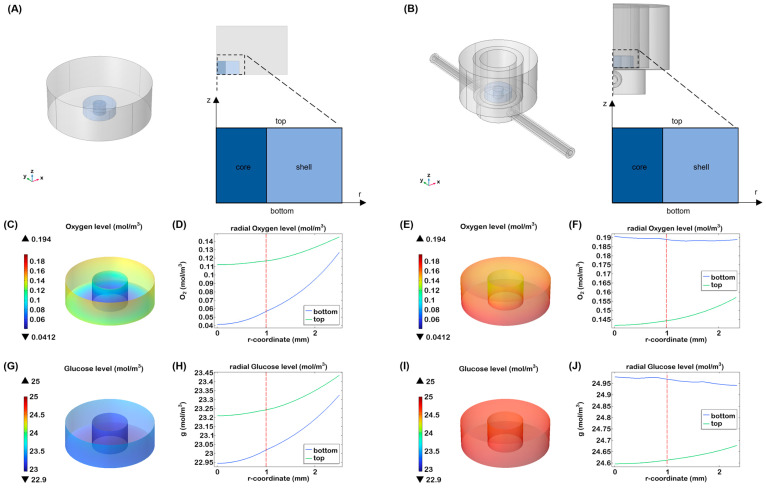
Nutrient transport in the core–shell hydrogel laden with breast cancer cells. Three-dimensional visualisation and cross-sectional view of (**A**) a 24 multi-well and (**B**) half of the MIVO device, with a zoom on the structure of the gel, indicating the two different compartments (core and shell), the top and the bottom surface; 3D visualisation of oxygen (**C**) and glucose (**G**) concentration in the heterogeneous hydrogel after 24 h of static culture; oxygen (**D**) and glucose (**H**) levels through the hydrogels at different heights after 24 h of culture; 3D visualisation of oxygen (**E**) and glucose (**I**) concentration in the heterogeneous hydrogel after 24 h of dynamic culture; oxygen (**F**) and glucose (**J**) levels through the hydrogels at different heights after 24 h of dynamic culture; the red line indicates the interface between the two zones.

**Figure 7 gels-11-00356-f007:**
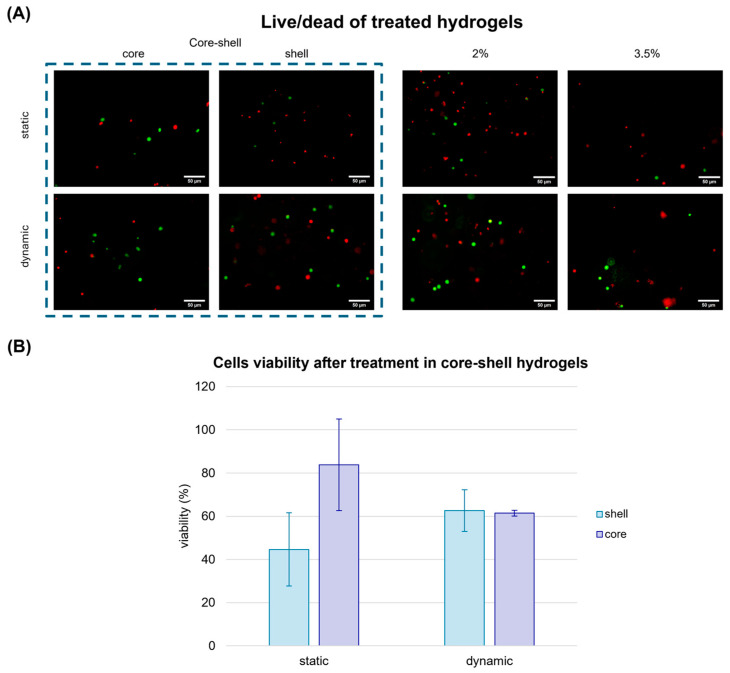
Cell viability assay performed in alginate hydrogels treated with cisplatin after one week of static culture and dynamic culture in MIVO organ-on-chip. (**A**) Live cells were marked with Calcein-AM, dead cells were marked with Propidium Iodide. Scale bar: 50 µm; (**B**) Cell viability evaluation in the hydrogels. The values are represented by mean ± std (*n* = 5), normalised with respect to the viability in the untreated gels. Statistics performed with two-way ANOVA; significance was assumed for *p* < 0.05.

**Figure 8 gels-11-00356-f008:**
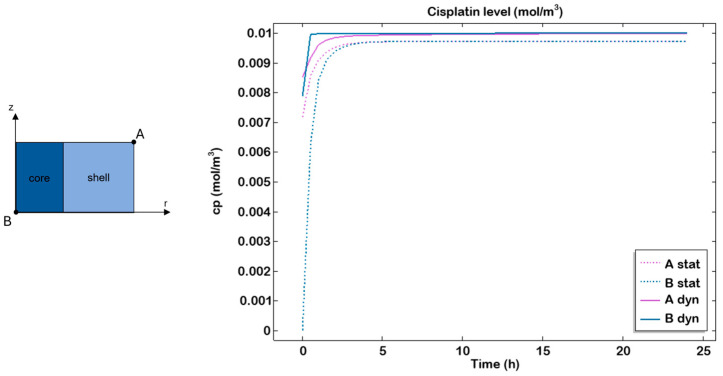
Cisplatin transport profile in static and dynamic conditions. Cisplatin transport within the hydrogel examined in the most external point at the top (A) and the most internal point at the bottom (B) in static (dashed line) and dynamic (continuous line) after 24 h of culture.

**Table 1 gels-11-00356-t001:** Young modulus of the different types of cell-laden hydrogels during culture time and cell-free hydrogels.

		Compressive Young Modulus (kPa)		
		2%	Core–Shell	3.5%
	day 1	7.13 ± 2.58	12.09 ± 2.18	21.14 ± 14.9
cell-laden	day 7	7.35 ± 2.94	16.24 ± 12.75	16.45 ± 7.01
	day 14	20.26 ± 6.15	23.68 ± 4.28	48.96 ± 9.14
cell-free		20.89 ± 6.96	44.69 ± 4.63	86.59 ± 17.34

**Table 2 gels-11-00356-t002:** Model Parameters and their Values.

Parameter	Value	Description	Reference
V_max,ox_	2.03 × 10^−16^ mol/s	Oxygen consumption rate for single cells	[54,55]
V_max,gl_	2.7 × 10^−16^ mol/s	Glucose consumption rate for single cells	assigned
V_max,cp_	1.66 × 10^−12^ mol/(m^3^/s)	Cisplatin consumption rate for unit of volume	[29]
K_m,ox_	6.875 × 10^−3^ mM	MM constant for oxygen	[54]
K_m,gl_	2.9 mM	MM constant for glucose	[43]
K_m,cp_	6.64 × 10^−3^ mM	MM constant for cisplatin	[29]
n	10^6^ cells/ml	Cell density	assigned
D_ox,med_	3.83 × 10^−9^ m^2^/s	Oxygen diffusivity in medium	[56]
D_gl,med_	6.2 × 10^−10^ m^2^/s	Glucose diffusivity in medium	[57]
D_cp,med_	1.304 × 10^−9^ m^2^/s	Cisplatin diffusivity in medium	[29]
D_ox,soft_	2 × 10^−9^ m^2^/s	Oxygen diffusivity in soft gel	assigned
D_gl,soft_	6 × 10^−10^ m^2^/s	Glucose diffusivity in soft gel	assigned
D_cp,soft_	0.86 × D_cp,med_	Cisplatin diffusivity in soft gel	[29]
Q	1 mL/min	Flow rate in OOC	assigned
D_ox,pipe_	6.67 × 10^−11^ m^2^/s	Oxygen diffusivity in tubes	[58]

## Data Availability

The original contributions presented in this study are included in the article. Further inquiries can be directed to the corresponding author.

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
