# Peer review of "Core-Shell Hydrogels with Tunable Stiffness for Breast Cancer Tissue Modelling in an Organ-on-Chip System"

_gels, 2025, doi:10.3390/gels11050356_

Round 1
Reviewer 1 Report
Comments and Suggestions for Authors
31-32
These fundings suggest that integrating a compartmentalized core-shell cell?
correct the sentence.
180
The phrase "Cells displayed a round morphology and a homogeneous spatial distribution" suggests no significant difference between shell and core.
However, later, the text states "Notably, immunofluorescence revealed colocalization of Ki67 and HIF-1α, particularly in the stiffer core regions", indicating some spatial differentiation.
This slight contradiction could be clarified. Are the core and shell truly homogeneous, or do they exhibit functional differences despite similar morphology?
184-186
The paragraph suggests that the stiffer core regions exhibit more colocalization of Ki67 (proliferation marker) and HIF-1α (hypoxia marker). However, it is not explicitly stated whether hypoxia is promoting proliferation or whether cells are merely adapting to it.
Consider discussing whether hypoxia-induced proliferation is expected in this model and how it relates to in vivo tumor behavior.
209-212
The paragraph states that nutrient diffusion under dynamic conditions improves cell viability in the stiffer core region, but it does not address whether the increased viability is linked to changes in proliferation or metabolic adaptation.
Consider adding data or discussion on whether cells in the dynamic condition also show increased proliferation or metabolic activity in addition to improved viability.
243–244
The text states that glucose concentrations did not decrease significantly over time in static conditions. It later mentions that under dynamic conditions, glucose did not show a significant gradient (lines 250–251) but was slightly higher at the bottom of the hydrogel (line 252).
However, the reasoning for this slight increase at the bottom is vague—fluid flow might enhance delivery, but wouldn't it also promote more uniform distribution rather than localized enrichment?
Comments on the Quality of English Language
The manuscript is well-structured and conveys the scientific findings clearly. However, there are several grammatical errors and inconsistent word choices that could be improved for better readability and professionalism.
Reviewer 2 Report
Comments and Suggestions for Authors
This study reports the preparation of core-shell alginate hydrogels with stiffness gradient to mimic the mechanical heterogeneity and hypoxic conditions for breast cancer modelling. Integrated with a MIVO® organ-on-chip, it replicates fluid flow and improves drug diffusion. Cisplatin tests reveal stiffness-dependent responses, highlighting the model’s potential for physiologically relevant cancer studies. This study is well designed and the methodology is sound. The writing is clear and conclusions are supported by the data. Overall, this is a high-quality work and I would be happy to recommend it for publication. There are some minor formatting issues as follows.
Line 21: “a good cells proliferation” could be “good cell proliferation”
Line 33: “These fundings suggest...” could be “These findings suggest...”
Line 197: “diplays” could be “displays”
Line 271: “are not a fully humanized model” could be “are not fully humanized models”
Reviewer 3 Report
Comments and Suggestions for Authors
Comments and Suggestions for Authors
The manuscript by Parodi et al. focuses on the development of core-shell hydrogels for breast cancer tissue modelling. The topic is highly interesting and potentially of significant scientific relevance and utility for researchers working in this area. However, in my opinion, the manuscript contains several points that need to be clarified or expanded upon before it can be considered for publication. Specifically:
- The authors are encouraged to provide a more detailed comparison between their core-shell hydrogel model and other existing 3D tumor models that incorporate stiffness gradients, particularly in terms of mechanical fidelity and biological relevance.
- Given that different alginate concentrations were used in hydrogel formulation, the authors should clarify whether this variation also influenced gelation times. Could the authors report the gelation times for each concentration used?
- Are there any available data on the thixotropic behavior or injectability of the hydrogels? Such properties may have implications for future translational or in vivo applications.
- Was the long-term stability of the core-shell hydrogels evaluated? If so, the authors should report any findings related to structural integrity and functional performance over time.
- The manuscript mentions the possibility of cell migration out of the hydrogel. Were any preliminary observations or quantitative data collected regarding this behavior? How might these findings influence the interpretation of metastatic potential within this model?
- Could the authors justify the use of a 0.1 M CaClâ‚‚ solution embedded in the agar mold rather than directly mixing it with the alginate solution? A discussion on the implications of this choice for cross-linking uniformity and reproducibility is recommended.
- Was any pharmacokinetic modeling performed to simulate drug exposure profiles or concentration gradients of cisplatin within the hydrogel under flow conditions? Such modeling would enhance the physiological relevance of the system.
- Were nutrient depletion or metabolite accumulation monitored throughout the 7-day culture period? These factors could significantly influence cellular responses independent of drug treatment.
- For better clarity and comparison, the authors are requested to summarize the mechanical characterization data presented in Figure 3 into a concise and well-structured table.
- Regarding the immunofluorescence and cell viability experiments, it would be beneficial to include corresponding bright-field images to provide spatial context and aid in result interpretation.
- The authors should consider expanding the literature review to include additional relevant studies in the field. A more thorough discussion of previously published tumor models would better contextualize the novelty and impact of the current work.
Round 2
Reviewer 3 Report
Comments and Suggestions for Authors
The authors have conducted an extensive and well-elaborated revision. In my opinion, the manuscript is now more concise and scientifically relevant. However, a few minor comments remain:
-
In Table 1, in addition to the Young’s modulus values, would it be possible to include the values of G′ (storage modulus) and G″ (loss modulus) for the gels? Were these measurements acquired?
-
Regarding the immunofluorescence and cell viability experiments, bright-field images would have been useful to better visualize not only cell morphology but also spatial localization. These could be included as supplementary material to support the fluorescence data. However, I understand that such analyses were not performed in the present study. Therefore, I would expect them to be included in future experiments or data acquisitions.
-
The authors have expanded the number of cited manuscripts relevant to the topic, as requested. However, in my view, the number of references remains limited. I suggest including citations to the following articles in the Introduction:
-
10.1016/j.ijpharm.2020.119219
-
10.3390/pharmaceutics15031026
-
10.7150/thno.90093. PMID: 38164155
Aside from these remarks, I have no further requests. In my opinion, the manuscript is now in a more complete form and suitable for publication.
